# Importance of Probiotics in Fish Aquaculture: Towards the Identification and Design of Novel Probiotics

**DOI:** 10.3390/microorganisms12030626

**Published:** 2024-03-21

**Authors:** Edgar Torres-Maravilla, Mick Parra, Kevin Maisey, Rodrigo A. Vargas, Alejandro Cabezas-Cruz, Alex Gonzalez, Mario Tello, Luis G. Bermúdez-Humarán

**Affiliations:** 1Facultad de Medicina Mexicali, Universidad Autónoma de Baja California, Mexicali 21000, Mexico; edgar.torres.maravilla@uabc.edu.mx; 2Laboratorio de Metagenómica Bacteriana, Centro de Biotecnología Acuícola, Universidad de Santiago de Chile, Santiago 9160000, Chile; mick.parra@usach.cl (M.P.); rodrigo.vargas.c@usach.cl (R.A.V.); 3Laboratorio de Immunología Comparativa, Centro de Biotecnología Acuícola, Universidad de Santiago de Chile, Santiago 9160000, Chile; kevin.maisey@usach.cl; 4Unidad de Producción Acuícola, Universidad de Los Lagos, Osorno 5290000, Chile; 5Anses, INRAE, Ecole Nationale Vétérinaire d’Alfort, UMR BIPAR, Laboratoire de Santé Animale, F-94700 Mai-sons-Alfort, France; alejandro.cabezas@vet-alfort.fr; 6Laboratorio de Microbiología Ambiental y Extremófilos, Departamento de Ciencias Biológicas y Biodiversidad, Universidad de Los Lagos, Osorno 5290000, Chile; alex.gonzalez@ulagos.cl; 7Micalis Institute, Université Paris-Saclay, INRAE, AgroParisTech, 78350 Jouy-en-Josas, France

**Keywords:** pathogen, fish immunology, probiotic, recombinant probiotic, microbiota

## Abstract

Aquaculture is a growing industry worldwide, but it faces challenges related to animal health. These challenges include infections by parasites, bacteria, and viral pathogens. These harmful pathogens have devastating effects on the industry, despite efforts to control them through vaccination and antimicrobial treatments. Unfortunately, these measures have proven insufficient to address the sanitary problems, resulting in greater environmental impact due to the excessive use of antimicrobials. In recent years, probiotics have emerged as a promising solution to enhance the performance of the immune system against parasitic, bacterial, and viral pathogens in various species, including mammals, birds, and fish. Some probiotics have been genetically engineered to express and deliver immunomodulatory molecules. These promote selective therapeutic effects and specific immunization against specific pathogens. This review aims to summarize recent research on the use of probiotics in fish aquaculture, with a particular emphasis on genetically modified probiotics. In particular, we focus on the advantages of using these microorganisms and highlight the main barriers hindering their widespread application in the aquaculture industry.

## 1. Introduction

The United Nations estimates that the human population will reach 9.7 billion by 2050 and 10.4 billion by 2100 [1]. Population growth and consequent climate change have generated concerns about food security and its sustainability [2]. Aquaculture has been presented as a means of obtaining food with excellent protein, vitamin, and mineral values [3]. Aquaculture has been strengthened due to continued growth, the increase in freshwater value chains, research in genetics and nutrition, diversification of nutrition sources to reduce the use of fishmeal, and the increase in the use of bivalves and seaweed to bolster the production chain and sustainability [4]. An example of the diversification of aquaculture is that, in 1950, just 73 species were used in farming, whilst in 2017 this number had risen to 415 [5].

According to the Food and Agriculture Organization (FAO) in its 2020 report, world fish production was 179 million tons (Mton), with an estimated value of 401 billion USD. Of total production, aquaculture represented 46% with a total of 82.1 Mton, where finfish (53.2 Mton), molluscs (17.7 Mton) and crustaceans (9.4 Mton) predominate. Growth projections suggest that, by 2030, the contribution of aquaculture to total production will be 53%, with the main producing countries being China, Bangladesh, Chile, Egypt, India, Indonesia, Norway, and Vietnam. The main farmed fish species include grass carp (*Ctenopharyngodon idellus*), silver carp (*Hypophthalmichthys molitrix*), Nile tilapia (*Oreochromis niloticus*), common carp (*Cyprinus carpio*), bighead carp (*Hypophthalmichthys nobilis*), catla (*Catla catla*), crucian carp (*Carassius* sp.), and Atlantic salmon (*Salmo salar*) [6]. However, as happens with food grown on land, aquaculture faces new challenges to the continuing of its expansion, especially with regard to climatic externalities that affect its entire development chain, as well as difficulties inherent to intensive farming [6].

During the development of aquaculture, one of the key problems has been the incidence of pathogens. The most prevalent ways to combat infections have been chemical agents and antibiotics [7]. However, in recent years the use of antibiotic treatments has been questioned and restricted in several countries, due to their bio-accumulative effects and the increase in bacterial antibiotic resistance with detrimental consequences on human and animal health [8,9]. Some of the alternatives to the excessive use of antibiotics in order to deal with pathogens in aquaculture are vaccination and the use of probiotic strains (Figure 1). Indeed, the food and agriculture organization of the united nations (FAO) and the World Health Organization (WHO) have defined probiotics as “live microorganisms, which, when administered in adequate doses, confer benefits to the health of the host” [10]. Taking into account the variations in terrestrial and aquatic animal life, Merrifield et al. 2010 introduced a variation of the term probiotics with a focus on aquaculture, defining a probiotic as “an organism that can be considered alive, dead or a component of a microbial cell, which administered via feed or rearing water, benefiting the host by improving disease resistance, health status, growth performance, feed utilization, stress response, or general vigor, which is achieved at least in part by improving the microbial balance of the host or the microbial balance of the environmental setting” [11].

Among the bacterial genera commonly utilized as probiotic microorganisms in aquaculture practices, prominent groups include lactic acid bacteria (LAB), *Bacillus*, *Alteromonas*, *Arthrobacter*, *Bifidobacterium*, *Clostridium*, *Paenibacillus*, *Phaeobacter*, *Pseudoalteromonas*, *Pseudomonas*, *Rhodosporidium*, *Roseobacter*, and *Streptomyces* [12]. Additionally, eukaryotic microorganisms such as microalgae (*Tetraselmis*) and yeasts from genera *Debaryomyces*, *Phaffia*, and *Saccharomyces* have demonstrated efficacy in probiotic assessments [12]. Furthermore, certain isolates from the pathogenic genera *Aeromonas* and *Vibrio* exhibit probiotic properties [13,14].

The main methods by which probiotics generate defense against diseases are the modulation of immune parameters, competition for binding sites, production of antibacterial substances, and competition for nutrients [15]. Excellent reviews of traditional probiotics used in aquaculture have been published in recent years [15,16,17,18]. The focus of this review is to describe the impact of probiotics and their metabolites on the treatment of the main pathogens that affect fish aquaculture, and how recombinant probiotics act as specific alternative or complementary biopharmaceuticals to antibiotic treatment.

## 2. Probiotics

### 2.1. Probiotics in Aquaculture

Global aquaculture production has nearly doubled every ten years, demonstrating the significant and growing role of fisheries and aquaculture in providing food, nutrition, and employment. At the global level, since 2016, aquaculture has been the main source of aquatic animals available for human consumption. In 2020, this share was 56%, a figure that can be expected to continue to increase in the long term [19]. In the same year, fish (finfish) accounted for 76% of the total aquatic animals produced through aquaculture [19]. To improve nutrition and food security, sustainable development of the industry requires advanced disease and health management because the aquatic environment renders fish particularly susceptible to ubiquitous pathogens. For many years, antibiotics were used for pathogen control in the fish farm sector, leading to antibiotic resistance and negative consequences for animal and human health. Therefore, probiotic strains, “live microorganisms that, when administered in adequate amounts, confer health benefits” [10], have emerged as new alternatives to therapeutic and prophylaxis treatments in ensuring nutrition, food security and sustainable development of the industry. Several surveys of gut bacterial communities agree that the fish gastrointestinal tract (GIT) harbors a bacterial load of approximately 10^8^ bacterial cells per gram. The fish gut microbiota is dominated by Proteobacteria (51.7%) and Firmicutes (13.5%), different from the dominant taxa reported in terrestrial vertebrates (Firmicutes and Bacteroidetes) [20]. Among fish, herbivores harbor the most diverse microbiomes because they require bacteria, such as *Clostridium*, *Leptotrichia*, or *Citrobacter*, to digest plant-derived cellulose [21]. The aquaculture sector has employed strategies to enhance fish production by using probiotic strains and/or compounds that stimulate their microbiota (Table 1). In the early stages of production of farmed fish, fish require supplementation of live feed, which can introduce pathogens into the closed system. However, this can be handled by the introduction of probiotic strains, which can also help in the degradation of indigestible compounds for fish in larval stages. Probiotics can increase growth performance and digestive enzyme activity (i.e., lipase, protease, and amylase activities) [22]. For instance, when larval pike-perch (*Sander lucioperca*) diets were supplemented with *Lactobacillus paracasei* BGHN14 + *Lactobacillus rhamnosus* BGT10, or with *Lactobacillus reuteri* BGGO6–55 + *Lactobacillus salivarius* BGHO1, improvements in skeletal development and the trypsin-to-chymotrypsin activity ratio, as an indicator of protein digestibility, were observed [23].

On the other hand, aquaculture procedures (handling, transport, or stocking density) activate the stress system, inducing negative effects on different physiological processes in fish (growth, reproduction and immunity). Administration of *Bacillus subtilis* and *Bacillus licheniformis* might be helpful in triggering metabolic advantages during stressful handling events on fish farms, as observed by reduced levels of cortisol tendency [24]. Moreover, administration of probiotic bacteria-derived purified cell components may eliminate problems associated with fish pathogens, such as *Aeromonas*, *Pseudomonas*, *Roseobacter*, and *Vibrio*. Cellular component of *B. subtilis* VSG1, *Pseudomonas aeruginosa* VSG2, and *Lactobacillus plantarum* VSG3 induced significantly higher lysozyme activity in rohu (*Labeo rohita*) [25], as did *B. licheniformis* in juvenile Nile tilapia [26]. Lysozyme is involved in innate immunity in fish, accounting for substantial antibacterial activity against both Gram-positive and Gram-negative bacteria, hydrolyzing the chemical bond between N-acetylmuramic acid and N-acetylglucosamine during bacterial cell wall degradation [26,27]. In addition, immunization with bacteria-derived purified cell components was found to induce interleukin IL-1β and Tumor Necrosis Factor alpha (TNF-α) expression, which is consistent with reports of their upregulation in response to dietary administration of secondary metabolites from *B. licheniformis* in other fish species [25,26]. While IL-1 β plays an important role in fish immunity by activating lymphocytes and phagocytic cells, and increasing resistance to *Aeromonas hydrophila* infection, the TNF-α family in fish exerts pro-apoptotic activity (as do its mammalian homologues) and upregulates granzyme expression in non-specific cytotoxic cells, protecting these cells from activation-induced cell death [28]. Similarly, *B. licheniformis* supplementation in diets increased the content of complement C3 in Nile tilapia serum [26].

While strains of bacteria used in aquaculture may be different from those used for human consumption, they can also provide health benefits and continue to be tested in aquaculture. For example, Muñoz-Atienza, et al. [29] screened classical LAB with in vitro antimicrobial activity, *Tenacibaculum maritimum* and *Vibrio splendidus*. In addition, the LAB *Enterococcus faecium* CV1, *E. faecium* LPP29, *Lactobacillus curvatus* subsp. *curvatus* BCS35, *Lactococcus lactis* subsp. cremoris SMF110, *Leuconostoc mesenteroides* subsp. cremoris SMM69, *Pediococcus pentosaceus* SMM73, *P. pentosaceus* TPP3, and *Weissella cibaria* P71 were found to be able to survive in seawater and resisted low pH and turbot bile. New approaches using high-throughput sequencing and gas chromatography/mass spectrometry metabolomics techniques have been used to identify beneficial microbes, such as *Undibacterium*, *Crenothrix*, and *Cetobacterium*, which were positively correlated with most intestinal metabolites in farmed Nile tilapia [30].

Another approach used to increase beneficial microbes in aquaculture is through prebiotic supplementation in fish diet formulations (see Table 1). Prebiotics are substrates that are selectively utilized by host microorganisms, conferring a health benefit [31]. Dietary citric and sorbic acid (organic acids, OA) and naturally identical compounds (NIC, specifically thymol and vanillin) were able to stimulate the development of beneficial bacteria taxa, such as *Lactobacillus*, *Leuconostoc*, and *Bacillus* spp., and decrease inflammation-promoting and homeostatic functions, as observed by dose-dependent up-regulation of IL-8, IL-10 and transforming growth factor-β (TGF-β). This study also identified a decrease in the putative genes encoding for protein, related to bacterial invasion of epithelial cells and bacterial toxins, in the microbiota of fish that received food supplemented with high doses of NIC and OA blend [32]. Baumgärtner, James, and Ellison (2022) demonstrated positive effects on beneficial bacterial taxa of the microbial community of the distal intestine and the skin of Atlantic salmon (*S. salar*), by a prebiotic mix of 1,3/1,6-β-glucans, mannan-oligosaccharides, nucleic acids, nucleotides, medium chain fatty acids and single chain fatty acids (SCFAs). The supplementation of a prebiotic improved the microbial community in the gut and the skin of Atlantic salmon, especially of *Bacillus* and *Mycoplasma* spp. species, as observed by 16S rRNA profiling [33].

Administration of dietary β-glucan provoked a prolonged effect on the fish innate immune function, and increased lysozyme activity in the plasma, liver, and intestines of Nile tilapia [34]. β-glucans can be used to enhance intestinal fish microbiota (i.e., Bacteroidetes) and produce derived compounds that stimulate immune responses. Petit et al. (2022) provide evidence of the ability of the intestinal microbiota of carp to ferment β-glucans, increase SFCA levels (acetate, butyrate, and propionate) in vitro, and regulate the expression *of gpr40L* genes (putative SCFA receptors) [35]. In addition, β-glucans combined with *Aspergillus oryzae* exert a “synbiotic effect” on growth, antioxidant, and immune responses (IgM, lysozyme activities) in Nile tilapia [36]. The use of synbiotics in fish is becoming increasingly relevant in aquaculture as a functional feed additive, given their abilities to enhance IgM, lysozyme, bactericidal, antioxidant, and phagocytosis activities, among others (see Table 1). Synbiotics can be defined as “a mixture comprising live microorganisms and substrate(s) selectively utilized by host microorganisms that confers a health benefit on the host” [37]. For example, dietary watermelon rind powder and *L. plantarum* CR1T5 introduced individually in the diet of Nile tilapia did not produce any significant effect. Nevertheless, in combination, they exert a synbiotic effect, as observed by stimulated growth, skin mucus, and serum immune parameters of Nile tilapia fingerlings, and significantly raised the relative percent survival and protection against *Streptococcus agalactiae* by 68% [38]. Prebiotic supplementation as an external carbon source in biofloc created higher floc formation, which is a valuable protein source for fish, and synergism between prebiotics and LAB generated a favorable intestinal environment leading to better nutrient utilization [38]. Additionally, Nile tilapia fed a combination of *Bacillus* NP5 strain and oligosaccharides from sweet potatoes displayed a better growth performance due to improved nutrient utilization and control of streptococci [39]. Furthermore, red tilapia fed with a synbiotic-supplemented diet (*L. rhamnosus* GG and *Helianthus tuberosus*) had a significantly increased goblet cell count (acid mucous cells, neutral mucous cells and double-staining mucous cells) in the proximal and distal intestine; indeed, the dominant mucous cells were of the acid type, which are those that are associated with protection against bacterial translocation [40]. Finally, the synergism between pistachio hull-derived polysaccharides and *Pediococcus acidilactici* improved antioxidant capacity (superoxide dismutase, catalase, and glutathione peroxidase) and immune-related genes (TNF-α, IL-1β, IL-10) in Nile tilapia and protected against *A. hydrophila* infection [22].

Summarizing, all these benefits are due to the fact that agricultural by-products are sources of natural antioxidants and dietary fibers, and they play a pivotal role in innate immunity and micro- and macro-nutrient absorption [38]. In addition, probiotics can interact with the immune cells of the GITs of fish, triggering immune responses in favor of fish development and protecting against pathogen infections. It is expected that the use of synbiotics will become a common alternative for the prevention and control of bacterial diseases in fish farms [40].

**Table 1 microorganisms-12-00626-t001:** Conventional probiotic strains, prebiotic and synbiotic combinations used in aquaculture.

Species/Size	Bacteria/Prebiotic	Pathogen (Challenge)	Oral Doses	Effects	Ref.
Pike-perch (*S. lucioperca*)/Juvenile (Larvae)	*L. paracasei* BGHN14 *L. rhamnosus* BGT10 *L. reuteri* BGGO6–55 *L. salivarius* BGHO1, OTOHIME fish diet *Artemia nauplii*		Fish diet: non-enriched *A. nauplii* per 5 day (300 nauplii per larvae per day) + 14 days of enriched diets (8 to 14 g per tank per day, 80 L/tank). Groups: non-enriched *A. nauplii* and OTOHIME hydrolyzed by BGHN14 + BGT10. *A. nauplii* enriched with BGHN14 + BGT10. *A. nauplii* enriched with BGGO6–55 + BGHO1. non-enriched *A. nauplii* and non-hydrolyzed OTOHIME.	↗ Better skeletal development. ↗ Higher trypsin to chymotrypsin activity ratio values. ↘ Lower levels of *Aeromonas* and *Mycobacterium* spp.	[23]
Turbot (*Scophthalmus maximus*)/95.8 ± 17.7 g	Yeast (*Saccharomyces cerevisiae*) β-glucan and mannan oligosaccharide (GM), Alginic acid (AC) from algal extracts containing 99% *Laminaria digitata* and 1% *Ascophyllum nodosum*. Purified yeast nucleotides (cytidine-5V-monophosphate (CMP)), disodium uridine-5V-monophosphate (UMP), adenosine-5V-monophosphate (AMP), disodium inosine-5V-monophosphate (IMP), disodium guanidine-5V-monophosphate (GMP)) and ribosomal RNA. *B. subtilis* and *B. licheniformis*		Fish diet: Hand fed twice daily for 84 days A basal low fish meal (FM; 32%) diet supplemented with: (i) Yeast (*S. cerevisiae*) β-glucan and mannan oligosaccharide (GM), (ii) Alginic acid (AC). (iii) Yeast nucleotides. (iv) Bacillus strains (BS), *B. subtilis* and *B. licheniformis*.	↘ Reduction of cholesterol levels. No changes in innate immune response. No changes in lysozyme activity in plasma.	[24]
Rohu (*L. rohita*)/ 43 ± 1.07 g	*B. subtilis* VSG1, *Pseudomonas aeruginosa* VSG2, and *L. plantarum* VSG3	*A. hydrophila*	Immunized intraperitoneally: 0.1 mL phosphate buffer solution (PBS) containing 0.1 mg of any of the following cellular components: intercellular products (ICPs) of *B. subtilis* VSG1, ICPs of *L. plantarum* VSG3, and heat-killed whole cell products of *P. aeruginosa* VSG2	↗ Intercellular products of *L. plantarum* VSG3. ↗ Higher post challenge relative percent survival (83.32%). ↗ Increase in ACP activity and induction of IL-1β and TNF-α expression.	[25]
Nile tilapia (*O. niloticus*)/3.83 ± 0.03 g	*B. licheniformis*	*Streptococcus* *iniae*	*B. licheniformis* (0%, 0.02%, 0.04%, 0.06%, 0.08% and 0.1% of AlCare^®^, containing live germ 2 × 10^10^ CFU/g)/twice daily fed for 10 weeks	↗ Improve the growth performance, enhance immunity by ↗ increasing the content of complement C3 in serum and lysozyme activity.	[26]
Turbot (*S. maximus* L.)/1.98 ± 0.17 g	*L. mesenteroides* subsp. cremoris SMM69 and *W. cibaria* P71	*V. splendidus* CECT528 *V. splendidus* ATCC25914 and *V. splendidus* DMC-1	Bathed with suspensions of bacteria at 1 × 10^9^ CFU/mL during 1 h at 18 °C twice: 0 and 24 h.	↗ Strong antimicrobial activity against *T. maritimum* and *V. splendidus*. Different adhesion ability to skin mucus. ↗ Inhibit the adhesion of turbot pathogens to mucus. ↗ Stimulation of genes encoding IL-1β, TNF-α, lysozyme, C3, MHC-Iα and MHC-IIα in five organs (head-kidney, spleen, liver, intestine and skin).	[29]
European Sea bass (*Dicentrarchus labrax*)/13.23 ± 0.18 g	Organic acids and natural identical compounds providing 25% citric acid, 16.7% sorbic acid, 1.7% thymol and 1% vanillin in a matrix of hydrogenated fats.		Feed was provided by hand/twice a day/6 days a week.	↗ Stimulation of the development of beneficial bacteria taxa such as *Lactobacillus*, *Leuconostoc*, and *Bacillus* spp. ↗ Dose-dependent upregulation of IL-8, IL-10 and TGF-β.	[32]
Atlantic salmon (*S. salar*)/~32 g	1,3/1,6-beta glucans, mannan-oligosaccharides, nucleic acids, nucleotides, medium chain fatty acids and single chain fatty acid.		Fed by hand 4 times/day, during 0, 6 and 12 weeks. Experimental blend containing prebiotics at 0, 0.5, 1, 2 g/kg in fish formulation.	Changes in gut and skin microbial community of salmon. ↗ Enrichment of *Bacillus* and *Mycoplasma* spp. species.	[33]
Nile tilapia (*O. niloticus*)/9.2 ± 0.1 g	β-glucans		Groups 1. 30 days of standard diet + 15 days of β-glucan. 2. 15 days of standard diet + 30 days of β-glucan diet. 3. 45 days of 0.1% β-glucan. Endpoint: 7 and 14 days post-feeding trial.	↗ Improvement of lysozyme activity in plasma, liver and intestine.	[34]
Nile tilapia (*O. niloticus*)/27.15 ± 0.2 g	*A. oryzae* and β-glucan		Fed 60 days 1. Standard diet 2. *A. oryzae* (1 g/kg) 3. β-glucan (1 g/kg) 4. 0.5 g/kg of *A. oryzae* + 0.5 g/kg of β-glucan	↗ Fish growth improvement ↗ Enhanced immune response by increase of IgM and lysozyme activities.	[36]
Nile tilapia (*O. niloticus*)/16.57 ± 0.14 g	Dietary watermelon rind powder (WMRP) and *L. plantarum* CR1T5 (LP)	*S. agalactiae*	Fish diets: 1. Standard diet 2. 40 g/kg of WMRP 3. 10^8^ CFU/g of LP 4. 40 g/kg of WMRP plus 10^8^ CFU/g of LP. Fish were hand-fed *ad libitum* twice daily during 8 weeks.	↗ Higher lysozyme and peroxidase elevation in skin mucus and serum. ↗ Phagocytosis and alternative complement (ACH50) activities. ↗ The relative percent survival of 68% in *S. agalactiae* challenge.	[38]
Nile tilapia (*O. niloticus*)/15–20 g	*Bacillus* subps. NP5	*S. agalactiae*	Fed 3 times/day/14 days before challenge. Diet: 1 g of probiotic (*Bacillus* NP5 at 1 × 10^6^ CFU/mL) and 2 g of prebiotic per 100 g of feed (oligosaccharides from sweet potatoes var. *sukuh*).	↗ Fish survival rate of 85.19% (control fed 18.52%). ↘ Level of damage by *S. agalactiae* in kidney and liver.	[39]
Red tilapia (*Oreochromis* spp.)/14.05 ± 0.42 g	Jerusalem artichoke (*H. tuberosus*) and *L. rhamnosus GG* (LGG)	*A. veronii*	Fish diet: Fish were hand-fed/twice day/30 days. 1. Standard diet. 2. 10 g/kg of Jerusalem artichoke (*H. tuberosus*) + 10^8^ CFU/g LGG).	↗ Growth performance by 106%. ↗ Enhanced blood glucose, total protein and total cholesterol levels. ↗ Enhanced intestinal parameters (villous height, absorptive area and globet cells) No changes of survival rate in *A. veronii* challenge.	[40]

↗ Indicates an increase, enhancement, or improvement, whereas ↘ denotes a decrease or reduction in the mentioned outputs.

### 2.2. Microbial Metabolites Produced by Probiotics and Intestinal Microbiota

Studies using mammalian models and zebrafish (*Danio rerio*) have shown that communication between microorganisms (probiotics or microbiota) and the host involves chemical cross-talk [41]. This communication involves interactions between host receptors/targets on immune cells and metabolites produced by microbial metabolism. This interaction alters the expression of immune genes, modifying the fate of some immune cells or the expression of cytokines [41,42].

Several metabolites produced by microorganisms have the ability to modify host cell metabolism and immune responses [43]. The SCFAs formate, acetate, n-propionate, n-butyrate and n-valerate are molecules produced by the fermentative anaerobic metabolism of bacteria belonging to the gut microbiota (mostly Clostridiales, from phylum Firmicutes). They are among the microbial molecules with the most significant impact on host physiology, reaching distant organs such as the brain due to their hydrophobic nature and low size, which enables them to be absorbed by intestinal epithelial cells and to diffuse through the host, producing effects in distal organs [44]. Butyrate is the most widely characterized microbial SCFA. It stimulates the extra-thymus production of Treg, PolyMorfoNuclear lymphocyte (PMN) activity, and the maturation and function of microglia [45,46], reduces the production of pro-inflammatory cytokines INF-γ, IL-1β, and TNF-α in macrophages [47], increases apoptosis and reduces the proliferation of T helper lymphocytes [48]. In dendritic cells, butyrate decreases the exposure of MHC-II, stimulating the production of anti-inflammatory cytokines (IL-22 and IL-10) [49,50]. In general, butyrate (and other SCFAs) produces an anti-inflammatory response; however, its precise effect depends on the SCFA and cell type. The wide spectra of effects related to butyrate can be explained by its capacity to stimulate the mammalian G protein-coupled receptor (GPCR), GPR41, GPR43, and GPR109a, beginning a cascade of phosphorylation mediated by ERK1/2 MAP kinase [51,52]. These receptors are differentially expressed in immune cells. For example, GPR43 is highly expressed in monocytes, macrophages/microglia, and neutrophils [45,47]. Butyrate also inhibits histone deacetylase 3 (HDAC3) involved in chromatin remodeling and produces epigenetic changes [53] that modify the cell fate of immune cells. In fish, butyrate has been identified in the gut of carnivorous and herbivorous specimens [54,55], promotes the expression of heat shock protein HSP70, pro-inflammatory factors (IL-1β and TNF-α), and anti-inflammatory cytokines (TGF-β) in *Cyprinus carpio* [56], and improves the inflammatory response in juvenile zebra fish [57]. Butyrate has been detected in the intestinal feces of Atlantic salmon at a concentration of around 1 mM [58,59]. Butyrate also has an immunostimulant activity when administered orally to Atlantic salmon, increasing the expression of mRNA encoding for C3 (complement marker) in head-kidney [60] by a mechanism currently unknown. Butyrate also inhibits the antiviral response in SHK-1 cells, inducing the expression of IL-10 and TGFβ in a mechanism independent of the expression of the butyrate receptor [58].

The intestinal microorganisms, and some probiotics, can metabolize the amino acid tryptophan (Trp) to produce indole-containing metabolites that regulate the immune system, activating the aryl hydrocarbon receptor (AHR) [61]. The bacteria responsible for this metabolism belong to the Firmicutes phylum, including members of the Lactobacillus, Clostridium, and Bacillus genera [62]. These metabolites stimulate the production of anti-inflammatory cytokines, promoting host–gut microbiota homeostasis [61]. Microbial indole-3-lactic acid (ILA) promotes the differentiation of CD4^+^ intraepithelial lymphocytes (IELs) into CD4/CD8 double-positive IELs [63]. Indole-3-acetic acid (IAA) and tryptamine (TRA) reduce the expression of inflammatory mediators, such as TNF-α and IL-1β, on monocytes/macrophages [64]. Indole-3-aldehyde (I3A) increases the expression of IL-22 in pancreatic innate lymphoid cells and promotes their differentiation toward regulatory macrophages and T-reg lymphocytes [65]. Indole-3-propionic acid (IPA), and indoxyl-3-sulfate (I3S) also regulate T cells and dendritic cells (DC) in the CNS [66]. Kynurenine (Kyn) and its derivates are also immune-active molecules that promote the apoptosis of Th1 cells, increasing the expression of IL-22 and a general anti-inflammatory response [65,67]. Some studies in other fish, such as Senegalese sole (*Solea senegalensis*), show that, in general, the administration of Trp improves the immune response by reducing the expression of inflammatory cytokines [68]. Trp can mitigate cannibalism, improve the growth of Asian Sea Bass (*Lates calcarifer*) [69], and counteract the effects of acute stress in Atlantic salmon [70]. In the case of Kyn, it has been described as a pheromone in rainbow trout (*Oncorhynchus mykiss*); however, there are no reports associated with its function as an immunomodulator [71]. Recent metabolomics studies have identified the presence of ILA, IAA, TRA, Kyn and Trp in the intestinal feces of Atlantic salmon [72].

In addition to Trp metabolites, microorganisms can also produce or activate neurotransmitters such as dopamine, norepinephrine, serotonin, gamma-aminobutyric acids (GABA), acetylcholine, and histamine, which have direct effects on immune cells [73,74]. Receptors for dopamine are found in macrophages, dendritic cells, B lymphocytes, T lymphocytes, microglia, neutrophils, and NK cells. Dopamine has been shown to inhibit Treg cells [75] and/or promote their differentiation to Th2 cells [76]. Norepinephrine is recognized by adrenergic receptors (alpha and beta), which are present in various immune cells. In peripheral tissues, norepinephrine interacts with dendritic cells, modifying the production of IL-10, IL-12, and IL-33, which in turn induce changes in naive T lymphocyte differentiation, modifying the balance among the T helper lymphocytes Th1, Th2, and Th17 [77]. Serotonin produces several immune effects, depending on its concentration and the type of serotonin receptor expressed on the immune cells [78]. Its production in intestinal enterochromaffin cells is stimulated by microbiota metabolites, such as SCFAs [79]. GABA produces different effects in the intestine depending on the cell type and the receptor; while in macrophages it promotes an inflammatory state with an increase in IL-1β, in dendritic cells and T lymphocytes it promotes an anti-inflammatory state with an increase in Treg cells [80]. Acetylcholine shows cell-dependent effects. Specifically, in macrophages it stimulates an inflammatory state through the production of IL-6, TNF-α, IFN-γ, and IL-12, while in T lymphocytes it promotes the formation of Treg cells [81]. Histamine shows pleiotropic effects that depend on the receptor stimulated. The histamine interaction with the H2R receptor results in an anti-inflammatory state that increases the production of IL-10 and inhibits the differentiation of T lymphocytes; however, its interaction with other histamine receptors produces an inflammatory stage increasing the production of IL-6, and IFN-γ, and promoting the differentiation of T lymphocytes to different T lymphocytes (Th1, Th2, and Th17) depending on the histamine receptor stimulated [82]. These neuro-immunomodulators are produced by several bacteria; for example, some bacterial strains from Lactobacillus or Pseudomonas genera can produce dopamine, norepinephrine, serotonin and histamine [74].

In aquaculture, few studies have analyzed the relationship between neurotransmitters and immunity. Most of the research has been performed on Rainbow trout and shows that serotonin and dopamine are increased in fish infected with *F. psychrophilum* [83]. Serotonin also reduces the proliferation of T lymphocytes [84] and acetylcholine reduces the expression of inflammatory cytokines in response to Poly I:C [85].

### 2.3. Fish Microbiota and Natural Anti-α-Gal Antibodies Induced by Probiotics

Natural antibodies are a crucial component of the innate immune system. They are a type of immunoglobulin found in individuals who have not encountered specific pathogens [86]. These antibodies have been discovered in various vertebrates, including mammals [87], birds [88,89], reptiles [90] and fish [91,92]. They play a vital role in binding auto-antigens and exogenous antigens present on the surface of microbes such as fungi, viruses and bacteria [93].

Natural antibodies come in different isotypes, namely IgM, IgG, and IgA, and serve various functions, such as initiating apoptosis [94], activating complement [95], opsonizing antigens [96], and facilitating phagocytosis through FcR receptors [97], among others [86]. They typically bind to antigens shared by groups of pathogens [98], such as lipopolysaccharide, lipoteichoic acid, peptidoglycan [98], and the carbohydrate Galα1-3Gal (α-Gal) [99]. The oligosaccharide α-Gal has recently gained significant attention in the scientific community [100] due to its role as a major antigen responsible for protective immunity against α-Gal-expressing pathogens that affect humans (e.g., *Trypanosoma* spp. [101], *Leishmania* spp. [102], and *Plasmodium* spp. [103,104]), birds (i.e., *Aspergillus fumigatus* [105]), and fish (i.e., *Mycobacterium marinum* [106]).

In healthy humans, anti-α-Gal antibodies, specifically of the IgG, IgM, and IgA isotypes, are naturally produced as part of the immune response to continuous exposure to Gram-negative bacteria present in the gut flora. These bacteria express a wide range of α-Gal-linked glycans, primarily in Galα1,2-, Galα1,4-, and Galα1,6-R forms [107,108]. Pacheco et al. [107] recently provided evidence demonstrating the protective effect of probiotics with high α-Gal content against mycobacteriosis caused by *M. marinum*. They identified and isolated native *Aeromonas veronii* and *Pseudomonas entomophila* bacteria, both rich in α-Gal, from the gut of zebrafish. These bacteria were coated onto commercial feed and orally administered to the fish before an infectious challenge with *M. marinum* under controlled conditions. Zebrafish treated with each probiotic showed significantly higher levels of IgM antibody levels against α-Gal, and those treated with *P. entomophila* experienced a significant decrease in mycobacterial infection [107]. Previous studies have shown that anti-α-Gal IgM antibodies can block malaria transmission by mosquitoes in α-Gal-deficient mice [104]. These results suggest that natural anti-α-Gal IgM are a conserved component of the innate immunity in certain craniates, such as fish, birds, and humans.

Probiotic treatment in zebrafish was also associated with notable changes in the composition and abundance of the fish microbiota [107]. Furthermore, the abundance of some specific taxa showed a negative correlation with anti-α-Gal IgM levels, indicating a potential role of anti-α-Gal immunity in regulating the gut microbiota of fish, as reported in mammalian gut microbiota studies [109]. Interestingly, gene expression analysis in probiotic-treated fish challenged with *M. marinum* suggests that the protective mechanisms associated with anti-α-Gal immunity extend beyond the control of mycobacteria through anti-α-Gal antibody-mediated actions. These mechanisms may include B-cell maturation, induced innate immune responses, and positive effects on nutrient metabolism and oxidative stress [107]. The preliminary findings of this trial support the use of *A. veronii* and *P. entomophila* as probiotics against fish mycobacteriosis and emphasize the need for further research into α-Gal-mediated immunity in fish.

## 3. Recombinant Probiotics in Aquaculture

Recombinant probiotics represent the next generation of probiotics engineered to specifically produce an effect in the host, either by stimulation of the immune system or by modifying the microbiota composition or metabolism (Figure 2). The probiotics used as hosts are mainly LAB, such as *L. lactis*, *Lactobacillus casei*, *L. plantarum*, or other microorganisms with Generally Recognized as Safe (GRAS) status, including yeast, *B. subtilis* or *Escherichia coli* Nissle 1917 (reviewed in [110,111,112,113,114,115,116,117,118]). Several studies describe the immunomodulatory properties of expressing cytokines from hosts [110], or epitopes from pathogens, showing that these kinds of probiotics are a feasible therapeutic alternative to prevent diseases caused by parasites [119], or bacterial [120] and viral pathogens [121], as they stimulate the production of antibodies specific against pathogens or other microorganisms that share the epitope expressed by these probiotics. The expression of anti-inflammatory cytokines has also been used to treat intestinal immune diseases [122] or tumors [123] in animal models. Despite the abundant literature that supports the potential use of these probiotics, most studies have been developed to pre-clinical levels and, to date, none have advanced to phase III in clinical trials [124].

In aquaculture, the use of recombinant probiotics has been much less explored. In teleost fish, which have an immune system like that of mammals, such as humans or mice, the recombinant probiotics assessed have used microbial backgrounds of *L. lactis*, *L. casei* or *plantarum*, and recently *B. subtilis*. In salmonids, oral administration of *L. casei* species expressing epitopes from infectious pancreatic necrosis virus (IPNV) has been shown to confer protection against the virus [125,126,127,128,129], while *L. lactis* strains have been used to orally immunize against viral hemorrhagic septicemia virus (VHSV) [130]. *L. lactis* has also been used to immunize against hirame novirhabdovirus (HIRRV) in flounder (*Paralichthys olivaceus*) [131]. In *C. carpio* (common carp), oral administration of recombinant *L. casei* expressing epitopes of *A. veronii* [132,133,134,135] or *A. hydrophila* confers protection against these pathogens [136], while the administration of *L. plantarum* expressing G protein of spring viremia of carp virus (SVCV) [137] and the ORF81 protein of koi herpesvirus (KHV) grants protection against both viruses in challenge assays, with high titers of IgM after its oral administration to *C. carpio* [138]. *L. lactis* has also been successful in triggering immunization against SVCV in *C. carpio* [139]. In crucian carps (*Carassius carassius*), an increment in the survival after challenge assays with *A. veronii*, *Vibrio mimmicus*, or *A. hydrophila* has been observed after the oral administration of *L. casei* expressing OmpAI from *A. veronii* [140], OmpK from *Vibrio mimicus* [141] or *L. plantarum* expressing Maltoporin from *A. hydrophila* [142], respectively.

In Nile tilapia, recombinant probiotics belonging to the *Lactococcus* and *Bacillus* genus have been employed. For instance, *L. lactis* has been used to express epitopes from *S. agalactiae*, increasing its survival in challenge assays after the oral administration of this probiotic [143].

Besides the immunization against bacterial and viral pathogens in fish, recombinant probiotics can also confer immunization against protozoa, such as in the case of the oral administration of *L. plantarum* expressing epitopes from *Ichthyophthirius multifiliis* in goldfish (*Carassius auratus*) [144].

Recombinant probiotics have also been used to express proteins that stimulate the immune response in fish, such as cytokines [145,146] and chemokines [147], the intestinal barrier [148], or enzymes that disrupt chemical communication in pathogens [149]. In the case of the cytokines, *L. lactis* has been used to deliver Interferon I and II, conferring protection against IPNV and *F. psychrophilum*, respectively [145,146]. On the other hand, a strain from the *Bacillus* genus isolated from the intestinal microbiota of Nile tilapia has been used to express CC-Chemokine, increasing the humoral and cellular immune response in Nile tilapia [147]. In the case of enzymes that disrupt the communication between bacterial pathogen cells, *B. subtilis* has been used to express the AiiO-AIO6 lactonase that hydrolyzes homoserine lactone (HSL), the molecule responsible for quorum sensing in *A. veronii* and several other Gram-negative pathogens. Their oral administration to zebrafish infected with *A. veronii* reduced the intestinal damage and the invasiveness of *A. veronii*, improving the survival rate after infection [149].

Altogether, these results have shown that LAB are an efficient vehicle for the release of immunostimulant peptides in fish. As mentioned above, the main strategy implemented the use of LAB to express epitopes from microbial pathogens. To achieve this goal these studies have cloned genes expressed on the surface of the pathogen, such as VP2-VP3 from IPNV [125,126,127,128,129], glycoprotein from HIRRV [131], SVCV [138,139], and VHSV [130], outer membrane proteins from *A. veronii* [132,133,140], or surface immunoreactive proteins from *S. agalactiae* [143] or *I. multifiliis* [144], under inducible promoters that respond to xylose (Pxyl) or nisin (Pnis) (Table 2). These genes were modified to achieve protein accumulation on the surface of the LAB by introducing signal secretion peptides such as Usp45 [150] or ssUSP [132,133,140] at their N-terminal in the case of recombinant probiotics that used *L. lactis* or *Lactobacillus* as host, respectively. These genes were also modified to improve the adherence of the encoded protein to the surface of the LAB or the epithelial host cells. The binding of the protein to the bacterial surface was achieved by introducing the C-terminal cell wall attachment (CWA) domains present in the protein encoded by *pgs*A [125,126,132,133,140], *acm*A [131,139] or *emm*6 [138]. The oral administration of these probiotics either, in the feed or by intubation every 3 days, was enough to induce the presence of IgM in serum mucosa in Nile tilapia and rainbow trout 4 days post-immunization [125,126,127,128,129,130,142]. The level of antibodies increased in the case of a booster applied in most cases 30 days after the first immunization [128,129,130,131,132,133,134,135,136,137,138,140,141] (Table 2). The fusion of the antigen to a CWA domain did not increase the level of serum antibodies with respect to the construction without the CWA domain. The localization of these antigens on the surface of the bacteria was checked using fluorescent antibodies against the antigen, which bind only to cells that express the fusion of the antigen with the CWA domain. The antibodies produced as a consequence of the immunization using these recombinant probiotics that expressed antigens from viral pathogens were effective in neutralizing in vitro infection, thus reducing the load of the pathogens and increasing survival in challenge assays, reaching in some cases double that of fish fed with the probiotic without the expression of the viral antigen (Table 2).

When the immuno-stimulation properties of some of these recombinant probiotics were evaluated using molecular markers, the probiotics expressing antigens from viral pathogens were able to induce an inflammatory response with an increment in the expression of IL-1β and TNF-α in the immune organ (spleen) and head-kidney, but also induced the expression of IFN-α, IFNγ and IgG [127,128,139], suggesting activation of the TH2 response, in agreement with the increment in the serum antibodies. The increment of these markers was higher than the induction observed in fish fed with probiotics containing the empty vector, which suggests that this stimulation is a consequence of the expression of viral antigens. A similar behavior was observed when Lactobacillus was used to express antigens from the parasite *I. multifiliis* [144] or the bacterial pathogen *A. veronii*. In the case of *L. plantarum* NC8 expressing the IAG-52X antigen from *I. multifiliis*, oral administration for four weeks was enough to induce the expression of C3, IgM, and MHC-I and increase survival from 40% to 60% in challenge assays [144]. *L. casei* CC16 expressing outer membrane protein of *A. veronii* TH0426 [132,133,134,135,140] or *A. hydrophila* [136,142] was also able to stimulate immune responses, increasing lysozyme activity, alkaline and acid phosphatases, and superoxide dismutase activity in serum, which suggests stimulation of the innate immune response. This stimulation was also associated with an induction of the expression of IL-1β and IFN-γ in the spleen, and TNF-α in the head-kidney.

A different strategy of immunomodulation based on the expression of interferon has been published recently. In this study, the oral administration of recombinant *L. lactis* producing Interferon Ia induced the expression of Mx and PKR in immune organs and also produced a reduction in the viral load in fish treated with these probiotics and challenged with IPNV [146]. A similar result using *L. lactis* expressing Interferon gamma activated the cascade of response to IFN-γ in immune organs, producing an increase in serum lysozyme activity after the end of the administration. This stimulation yields an increment in the survival to *F. psychrophilum* challenges, suggesting a stimulation of the innate immune response mediated by IFN-γ, since the challenge was initiated 7 days after the end of treatment with the recombinant probiotics. This result was not observed when the fish were fed only with the *L. lactis* strain without the modification [145]. A similar approach was used in Nile tilapia where a Bacillus strain isolated from the intestinal tract of this fish was modified to express a CC-Chemokine of this fish. The oral administration of the recombinant strain stimulated the humoral and cellular immune responses [147].

An interesting approach that combines immuno-stimulation by antigens and cytokines was published by Liu’s team. In this research, the VP2 protein of IPNV was expressed by fusion to CK6 chemokine which promotes macrophage/lymphocyte translocation. The probiotics expressing the fused peptides showed better results in stimulating the immune response inducing IL-8, Mx, MHC-II, and CK6 which resulted in an increment of titer of serum antibodies specific for VP2, an increased titer of neutralizing antibodies, and a greater capacity to reduce the viral load of IPNV in challenge assays [127,128]. This work strongly suggests that a new kind of adjuvant could be developed combining the stimulatory properties of some cytokines and antigens, both secreted by probiotics.

The reviewed studies employ a strategy of introducing genes into plasmids capable of replicating in LAB to modify their introduction. The primary plasmids utilized are derivatives of pNZ8148, pG, pSIP, pYG, and pNZ8149. However, all plasmids except pNZ8149 utilize antibiotics as selectable markers, which hinders their commercial use due to the introduction of antibiotic resistance genes into aquatic environments. This implies that a wide variety of plasmids or vectors must be designed to avoid the use of non-selectable markers that confer resistance to antimicrobials. Instead, metabolic selectable markers that provide the ability to metabolize certain nutrients should be employed, such as pNZ8149 in *L. lactis* NZ3900.

The choice of host strain is also an important aspect to consider, as most of the reviewed studies are based on conventional LAB isolated from terrestrial environments. These LAB strains exhibit weak colonization in the fish gut, with only approximately 1% retention observed after one week of administration. An exception was *L. casei* CC16, which was isolated from the intestine of common carp; in this case, retention was four times higher than that observed with LAB from other sources (4% vs. 1%). These results suggest that, for improved outcomes, recombinant probiotics should utilize host bacteria from the intestinal microbiota, promoting the development of autochthonous probiotics specific to each fish species. This approach diverges from the current generalized approach that employs the same probiotics for different species.

## 4. Future Perspectives of Recombinant Probiotics in Aquaculture

The development of sustainable aquaculture that generates a high-quality protein at a low cost is one of the challenges facing humanity in the coming years in order to sustain the growing global population, with minimal impact on terrestrial and oceanic environments. To achieve this objective, aquaculture faces a series of obstacles, including sanitary challenges, such as outbreaks of viral and bacterial pathogens that find ideal conditions in intensive aquaculture to spread and cause mortality. The strategies used in mammals to combat these outbreaks, such as the use of antibiotics and vaccines, have not shown the same utility in aquaculture production centers. Conventional vaccines, which are highly effective in mammals when administered by injection, do not demonstrate the same level of efficacy in fish, partly due to the difficulties and stress associated with individual vaccination of each fish and the differences between the immune systems of mammals and fish. Fish have a less developed acquired immune response compared to mammals. Antibiotics are effective against outbreaks of bacterial pathogens, but their application in open systems has an environmental impact, as they act as a selective factor for bacteria resistant to these antibiotics, which could potentially transfer this resistance to human pathogens.

In recent years, probiotics have emerged as an alternative. However, their non-specific effect on pathogens, coupled with the difficulties in isolating probiotic microorganisms from fish microbiota, as well as their incorporation into feed through extrusion processes carried out at high temperatures that reduce the number of viable cells, impose a barrier that has only been successfully overcome in aquaculture by probiotics such as *P. acidilactici* CNCM I-4622—MA 18/5M (Bactocell^®^, Lallemand, Montréal, QC, Canada) [151,152,153]. Recombinant probiotics, on the other hand, are emerging as a second alternative, as they can improve the properties of the original probiotics by expressing antigenic proteins of pathogens in such a way that they can confer immunity against the pathogens, stimulating the production of natural antibodies in the fish mucosa, as in α-Gal immunity. Recombinant probiotics offer a low-cost platform for expressing these immunogenic proteins, which would not need to be purified for use, and can be incorporated as probiotics in fish feed. So far, the use of recombinant probiotics as immunizing agents has only been tested with the pathogens that cause the greatest impact on aquaculture, leaving ample space for the development or study of their potential use in the preventive treatment of the majority of pathogens affecting aquaculture in various species.

Recombinant probiotics as vehicles for the expression of functional cytokines and chemokines have enormous potential for selective stimulation of the immune response of fish. They could act by (a) bypassing the inhibitions that pathogens exert on the immune system to achieve efficient infection, (b) generating prophylactic conditions that maintain a stimulated immune system capable of adequately controlling pathogens before they reach conditions or concentrations that hinder the action of the immune system, (c) enhancing the action of other immunostimulants or immunizing agents. To achieve these objectives, a greater understanding of the regulatory mechanisms of the immune system in each of the aquaculture species of interest is necessary. These species have evolutionarily distant immune systems from mammals, given their divergence over millions of years. Further studies are needed on the functioning of innate and acquired immune responses and the effects that cytokines or molecules secreted by microbiota microorganisms have on them, for the proper design of new probiotics with the ability to selectively stimulate the immune response.

From a market perspective, the development of platforms that eliminate the use of antibiotics and plasmids as vectors for protein expression in these recombinant probiotics is necessary, as various regulations prohibit the release of plasmids containing antibiotic-resistance genes into the environment. Although food-grade plasmids with metabolic markers for expression in *L. lactis* have been developed by [154], plasmids have the potential to be transferred among related microorganisms, even if they do not possess selection markers. For this reason, introducing recombinant probiotics into the genome through technologies like CRISPR would provide a stable expression platform with a very low transfer rate.

The identification of new probiotics, ideally from the GIT of each aquaculture species, which can be modified for the expression of immunostimulant proteins is also a challenge, as these microorganisms have a better colonization rate than probiotics from terrestrial environments. On the other hand, identifying a potential probiotic host that rapidly degrades within the ecosystem where aquaculture is conducted by ecosystem-specific microorganisms such as amoebas, would also help reduce the ecological impact caused by the introduction of these microorganisms in large quantities.

Finally, identifying probiotic strains capable of exerting their effects at low concentrations while achieving high yields (CFU/mL) during fermentation, and withstanding processing into fish feed, represents a desirable characteristic for reducing the production costs associated with probiotic-enriched fish feed. The expression of HSPs within these probiotic strains could potentially mitigate losses incurred during the inclusion process of probiotics into the feed [155,156].

## 5. Conclusions

Probiotics in fish aquaculture are a promising alternative to reduce the negative impact of pathogen outbreaks, reducing the economic losses produced by mortality of specimens, and the use of antibiotics applied to control the bacterial pathogen. Such factors will help make fish aquaculture a more environmentally friendly industry. Currently, most of the probiotics tested in fish aquaculture have been previously studied or applied in humans or mammals, making the development of new probiotics specialized for use in fish necessary. To achieve this goal, a better understanding of the mechanisms of interaction between fish intestinal microbiota and the host is necessary, characterizing the microbial metabolites involved that help to reduce the impact of the outbreaks, either by immunostimulant or antagonisms with the pathogens. Whole metagenomics studies could assist this characterization, allowing the identification of microorganisms without genes encoding for virulence factors, able to produce these microbial metabolites or those that encode for genes responsible for the synthesis of structural molecules with immunostimulant properties, such as α-Gal. Recombinant probiotics are other alternatives that allow the engineering of probiotics with specific immunostimulant, immunization, or metabolic properties, by the expression of genes that encode for these functions. These recombinant probiotics must be engineered using food-grade plasmids or ideally by the insertion of these genes in the chromosome of the bacterial probiotics without the presence of genes encoding for antibiotic resistance, using modern technologies of genetic engineering, such as CRISPR-CAS. However, to aid the proper design of recombinant probiotics, a better comprehension of the immune response of each species of fish produced in aquaculture against each pathogen (bacterial, fungal, or viral) is necessary in order to identify specific targets in the immune response of hosts to be stimulated or repressed. On the other hand, an improved understanding of the pathogenesis mechanism will allow the identification of targets in the pathogens to be selected as antigens to be over-expressed in the probiotics. These research lines must be accompanied by improvement in the technology employed to include these probiotics in the fish feed for a successful application in the fish aquaculture industry.

## Figures and Tables

**Figure 1 microorganisms-12-00626-f001:**
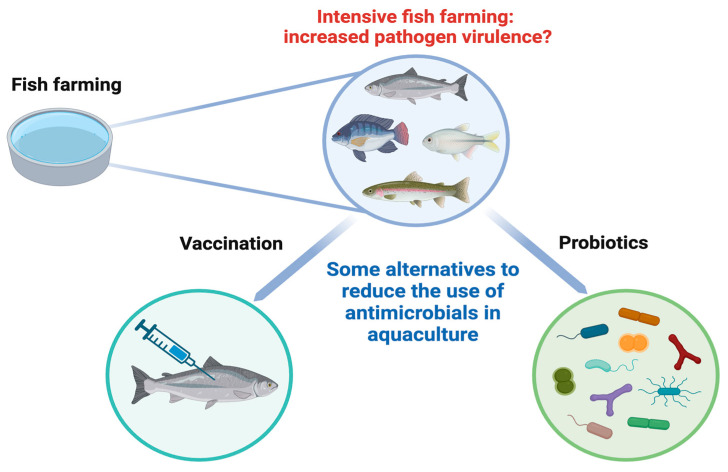
Use of probiotics as an innovative alternative to reduce the use of antibiotics in aquaculture.

**Figure 2 microorganisms-12-00626-f002:**
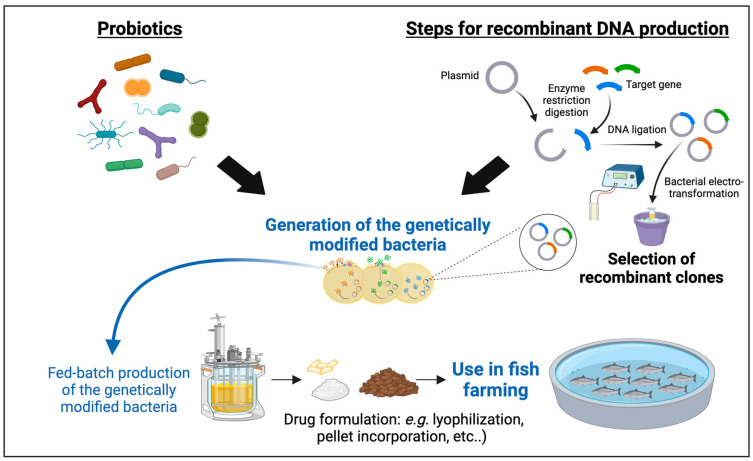
Use of genetically modified probiotics to fight some of the main infectious diseases in aquaculture.

**Table 2 microorganisms-12-00626-t002:** Recombinant probiotics tested in fish.

Species/Size	Bacteria	Vector ^(a,b,c)^	Immunostimulant Peptide	Pathogen	Oral Dosis	Effects	Ref.
Rainbow trout (*O. mykiss*)/100 g	*L. casei* ATCC 393	pG1-VP2 (Pxyl ^a^, ssUSP ^b^, *pgs*A ^c^), pG2-VP2 (Pxyl ^a^, ssUSP ^b^) pG1-VP3 (Pxyl ^a^, ssUSP ^b^, *pgs*A ^c^), pG2-VP3 (Pxyl ^a^, ssUSP ^b^)	VP2, VP3	IPNV	5 × 10^8^ CFU (once)	Anti-IPNV IgM increased 5 to 10-fold 31 days post immunization. Presence of neutralizing antibodies in serum 63 days post immunization. Up to 40-fold reduction of viral load in the spleen 10 days post-challenge. The challenge was performed on day 66 post-immunization.	[125]
Rainbow trout (*O. mykiss*)/100 g	*L. casei* ATCC 393	pG1-VP2-3 (Pxyl ^a^, ssUSP ^b^, *pgs*A ^c^), pG2-VP2-3 (Pxyl ^a^, ssUSP ^b^)	VP2-VP3 fusion	IPNV	5 × 10^8^ (once)	Presence of neutralizing antibodies in serum on day 63 post immunization. Anti-IPNV IgM increased 6- to 10-fold 31 days post immunization. Up to 10-fold reduction of viral load in the spleen 10 days post-challenge. The challenge was performed on day 66 post-immunization.	[126]
Rainbow trout (*O. mykiss*)/11.5 g	*L. casei* ATCC 393	pPG-612-CK6-VP2 (Constitutive ^a^, ssUSP ^b^)	CK6-VP2	IPNV	1 × 10^10^ (once)	CK6 expressed in Lactobacillus is biologically functional in vitro, increasing lymphocyte migration, inducing expression of IL-8, IL-1β and TNF-α. In vivo pPG-612-CK6-VP2 increase expression of IL-8, IL-1β, TNF-α, β-defensin, Mx, MHC-II, and CK6 in the first four days after administration. Increase in IgT and IgM titer by up to 10 times 31 days post immunization. Increase in neutralizing antibodies against IPNV.	[127]
Rainbow trout (*O. mykiss*)/10 g	*L. casei* ATCC 393	pPG-612-AHA1-CK6-VP2 (Pxyl ^a^, ssUSP ^b^)	AHA1-CK6-VP2	IPNV	2 × 10^9^ for 3 days, then booster on days 31, 32 and 33	AHA1-CK6 is biologically functional in vitro, increasing lymphocyte migration, inducing expression of IL-8, IL-1β and TNF-α. In vivo pPG- 612- AHA1-CK6-VP2 increase expression of IL-8, IL-1β, TNF-α, β-defensin, Mx, MHC-II, and CK6 in the first four days after administration. Increase in IgT and IgM titer by up to 15 times 31 days post immunization. Increase in neutralizing antibodies against IPNV. Reduced IPNV load.	[128]
Rainbow trout (*O. mykiss*)/15 g	*L. casei* ATCC 393	pPG-612-CK6-VP2-eGFP (Pxyl ^a^, ssUSP ^b^)	CK6-VP2-eGFP	IPNV	2 × 10^9^ for 3 days, then booster on days 31, 32 and 33	Increase in IgT and IgM titer by up to 5 times 15 days post primary immunization. Increase in neutralizing antibodies against IPNV. Reduced IPNV load	[129]
Rainbow trout (*O. mykiss*)/7 ± 0.65 g	*L. lactis* NZ3900	pNZ8148-G (pNZ8148 Pnis ^a^)	VHSV G	VHSV	Fed 3% daily. 10^8^ to 10^10^ CFU/g of feed for seven days and then boosted for one week in the third week	Induce IFN-α in the second week. Increase in IgM in serum after two weeks. Titers remain high until day 60. Reduced mortality by around 3-fold (from 60% to 20%). Reduced viral load in spleen and head-kidney. Increase the percent of weight gain (PWG) and reduced food conversion rate (FCR)	[130]
Olive flounder (*P. olivaceus*)/35 ± 5 g	*L. lactis* NZ9000	pSLC-G (pNZ8148, Pnis ^a^, SP-Usp45 ^b^, *acm*A ^c^)	HIRRV-G gene	HIRRV	1.0 × 10^9^ CFU/g diet, fed 1–2% each day. Supplemented food was administered for 7 days during weeks 1 and 5.	Increase in IgM titer against HIRRV in serum (after 4 weeks) and gut mucus (after 2 weeks). Serum IgM requires booster. Reduced viral load. Duplicated survival after challenge (70% vs. 35% in control).	[131]
Common carp (*C. carpio*)/56 ± 1 g	*L. casei* CC16 (Strain isolated from the common carp gut microbiota)	pPG1(Pxyl ^a^, ssUSP ^b^, *pgs*A ^c^) pPG2 (Pxyl ^a^, ssUSP ^b^)	OmpW	*A. veronii*	Fed daily at 1%. 1 × 10^9^ CFU/g of feed for three days, and then booster of another 3 days after two weeks	Increase in OmpW-specific IgM antibody two weeks post immunization. Increase in lysozyme, acid phosphatase, alkaline phosphatase, and superoxide dismutase activity in blood. Increase in phagocytic activity in serum. Induced expression of IL-1β, IL-10, IFN-γ, and TNF-α in spleen, head-kidney and gut. Increase in survival from 0 to 50% after challenge with *A. veroni* TH0426	[132]
Common carp (*C. carpio*)/50 ± 1 g	*L. casei* CC16 (Strain isolated from the common carp gut microbiota)	pPG1(Pxyl ^a^, ssUSP ^b^, *pgs*A ^c^) pPG2 (Pxyl ^a^, ssUSP ^b^)	OmpAI	*A. veronii* TH0426	Feeding rate 1% body weight. Immunization with 2 × 10^9^ CFU/g of feed for three days starting on day 1 and 31 (booster)	Increase in OmpAI-specific IgM antibodies in serum and skin mucose 15 days post immunization. Increase in lysozyme, acid phosphatase, alkaline phosphatase, and superoxide dismutase activity in blood after booster. Induced expression of IL-10, TNF-α in spleen, head-kidney and intestine. Induced expression of IL-1β, IFN-γ in spleen, head-kidney, gills, and intestine. Increase in survival from 0 to 50–70% after challenge with *A. veroni* TH0426	[133]
Common carp (*C. carpio*)/~60 g	*L. casei CC16* (Strain isolated from the common carp gut microbiota)	pPG-Aha1 (Pxyl ^a^, ssUSP ^b^, *pgs*A ^c^) pPG-Aha1-CTB (Pxyl ^a^, ssUSP ^b^, *pgs*A ^c^)	Aha1 CTB (Cholera toxin B-subunit) Aha-CTB	*A. veronii TH0426*	1 × 10^9^ CFU/g, days 1–3, 1st booster days 15–17, 2nd booster days 29–31. Challenge day 36	Recombinant strains stimulate IgM, acid phosphatase (ACP), alkaline phosphatase (AKP), C3, C4, lysozyme (LZM), Lectin and superoxide dismutase (SOD). Upregulate expression of: Interleukin-10 (IL-10), Interleukin-1β (IL-1β), Tumor Necrosis Factor-α (TNF-α), immunoglobulin Z1 (IgZ1) and immunoglobulin Z2 (IgZ2). Colonization of fish intestine. Confers protection against *A. veronii* infection; pPG-Aha1-CTB/Lc CC16 and pPG-Aha1/Lc CC16 shows relative percent survival (RPS) of 64.29% and 53.57%, respectively.	[134]
Common carp (*C. carpio*)/250 ± 2.5 g	*L. casei* CC16 (Strain isolated from the common carp gut microbiota)	pPG-Aha1, (Pxyl ^a^, ssUSP ^b^, *pgs*A ^c^) pPG-Aha1-LTB (Pxyl ^a^, ssUSP ^b^, *pgs*A ^c^)	Aha1 LTB (*E. coli* intolerant enterotoxin B subunit) Aha1-LTB	*A. veronii* TH0426	Carps were immunized orally by feeding fish food (2%) twice daily for three days, then booster at day 14.	Increase in specific IgM in serum, and in activities of ACP, AKP, SOD, LYS, C3, C4, and lectin. Increase in expression of IL-10, IL-1β, TNF-α, IgZ1, and IgZ2 in the liver, spleen, kidney, intestines, and gill tissues. Improved survival in fish challenged with *A. veronii* (60.71%).	[135]
Common carp (*C. carpio*)/50 ± 0.1 g	*L. casei* CC16 (Strain isolated from the common carp gut microbiota)	pPG1-Aha1 (Pxyl ^a^, ssUSP ^b^, *pgs*A ^c^) pPG2-Aha1 (Pxyl ^a^, ssUSP ^b^, *pgs*A ^c^)	Aha1 (*A. hydrophila*)	*A. hydrophila* BSK-10	Feeds containing 1 × 10^9^ CFU/g. The fish were orally immunized on day 1 to day 3, and reinforced posterior to 14 days (i.e., day 18–20).	Stimulate level of antibodies and AKP, ACP, SOD, LZM, C3, C4 in serum. Upregulate IL-10, IL-1β, TNF-α, IFN-γ in the livers, spleens, HK, and intestines. Increase in phagocytosis and survival rate (60–50%) after challenge with *A. hydrophila*.	[136]
Common carp (*C. carpio*)/200 ± 20 g	*L. plantarum*	pYG	SVCV-G	SVCV	Immunization with 1 × 10^9^ CFU/g of fed for three days on day 1, 10 (booster I) and 28 (booster II). Covered with alginate	Increase in anti-SVCV-G specific IgM antibodies in serum 14 days post primary immunization. Increase in survival from 0 to 80% in challenge assays. Increase in neutralizing antibodies	[137]
Common carp (*C. carpio*)/500 ± 50 g	*L. plantarum*	pYG-G (pYG301 derived, Pxyl ^a^, wall anchor motif from *Streptococcus pyogenes* M6 protein ^c^)	SVCV-G and KHV ORF81	SVCV KHV	Immunization with 3 × 10^9^ CFU/g of feed for three days on day 1, 14 (booster I) and 28 (booster II)	Increase in anti-SVCV-G IgM and anti-KHV-ORF81 IgM levels 14 days post primary immunization. Increase in neutralizing antibodies against SVCV and KHV. Reduced mortality caused by SVCV and KHV by 10% respect to fish fed with *L. plantarum*	[138]
Common carp (*C. carpio*)/5.05 ± 0.53 g	*L. lactis* NZ9000	pNZ-UGA (pNZ8148, Pnis ^a^, SP-Usp45 ^b^, *acm*A ^c^)	SVCV glycoprotein	SVCV	Intramuscular injection of 5 µg protein from culture	Induced IgM in serum 7 days post immunization. Induced TNF-α, IL-6b, IL-1β, Cxcr-1, IFN-γ, IFN-α and IgM. Increase in survival 8–9-fold. Reduced viral load	[139]
Crucian carp (*C. carassius*)/50 ± 1 g	*L. casei* CC16 (Strain isolated from the common carp gut microbiota)	pPG1(Pxyl ^a^, ssUSP ^b^, *pgs*A ^c^) pPG2 (Pxyl ^a^, ssUSP ^b^)	OmpAI-C5-I	*A. veronii* TH0426	Feeding rate 1% body weight. Immunization with 2 × 10^9^ CFU/g of fed for three days starting on day 1 and 31 (booster)	Increase in OmpAI-C5-I specific IgM antibodies in serum 16 days post immunization. Increase in lysozyme, acid phosphatase, alkaline phosphatase, and superoxide dismutase activity in blood after booster. Increase in phagocytic activity in serum. Induced expression of IL-10 in liver, spleen, kidney and intestine, Induced IL-1β, TNF-α, and IFN-γ in heart, liver spleen, kidney and intestine. Increased survival from 0 to 65–75% after challenge with A. veroni TH0426.	[140]
Goldfish (*C. auratus*)/50 ± 5 g	*L. casei ATCC393*	pPG-OmpK, (Pxyl ^a^, ssUSP ^b^, *pgs*A ^c^) pPG-OmpK-CTB (Pxyl ^a^, ssUSP ^b^, *pgs*A ^c^)	OmpK CTB (Cholera toxin B-subunit) OmpK-CTB	*V. mimicus* Hsy0531-k	10^8^ CFU/mL, mixed with commercial fish food First oral vaccination days 1–3, 2nd vaccination days 15–17, and 3rd vaccination days 29–31.	Lc-pPG-OmpK-CTB stimulated levels of IgM, and activity of acid phosphatase (ACP), alkaline phosphatase (AKP), superoxide dismutase (SOD), lysozyme (LYS), lectin, C3, and C4. Increase in expression of interleukin-1β (IL-1β), interleukin-10 (IL-10), tumor necrosis factor-α (TNF-α), and transforming growth factor-β (TGF-β) in the liver, spleen, head kidney, hind intestine and gills. Colonization of the intestine and increase in survival after challenge (58.33%).	[141]
Crucian carp (*C. carassius*)/65 ± 4 g	*L. plantarum* Lp-095	pPG-Malt-pgsA (Pxyl ^a^, ssUSP ^b^, *pgs*A ^c^)	Malt (Maltoporin)	*A. hydrophila*	Food supplemented with 10^9^ CFU/g. Fish were fed twice daily for 28 days without interruption.	Enhanced IgM level and phagocytic activity. Increase in expression of IL-10, IL-1β, TNF-α, IFN-γ in liver, spleen, head kidney and hind intestine. Increase in RPS of fish challenged intraperitoneal with *A. hydrophila* (55%).	[142]
Nile tilapia (*O. niloticus*)/15 ± 2 g	*L. lactis* NZ9000	pNZ8148-sip (pNZ8148 Pnis ^a^)	Surface immunogenicity protein (Sip)	*S. agalactiae*	2 × 10^8^–2 × 10^10^ CFU/fish	Increase in Sip specific IgM antibodies in serum 16 days post primary immunization. Increase in survival from 5 to 60% in challenge assays. Induced expression of IgT, IgM, CD8a and C3 in liver, spleen, intestine and thymus	[143]
Goldfish (*C. auratus*)	*L. plantarum* NC8	pSIP409-IAG-52X (pSIP409, Pspp ^a^)	IAG-52X	*I. multifiliis*	Fed 1% with 10^6^ CFU/g of feed, for 4 weeks	Increase in Ig in serum and skin after four weeks of feed. Increase in survival from 40 to 60% in challenge assays. Induced C3, IgM and MHC-I after 2 weeks of feed.	[144]
Rainbow trout (*O. mykiss*)/25 g	*L. lactis* NZ3900	pNZ8149 (Pnis ^a^, Usp45 ^b^)	Interferon II (Atlantic salmon)	*F. psychrophilum*	1 × 10^7^ CFU/fish each day for one week	Induced expression of IFN-γ, IP10, IL-6, STAT1 and IL-1β Increase in serum lysozyme activity Increase in survival from 50% to 80% in challenge assays	[145]
Atlantic salmon (*S. salar*)/10 g	*L. lactis* NZ3900	pNZ8149 (Pnis ^a^, Usp45 ^b^)	Interferon Ia (Atlantic salmon)	IPNV	1 × 10^7^ CFU/fish each day for one week	Induced expression of Mx and PKR in spleen and head kidney. Reduced viral load in spleen and head kidney	[146]
Nile tilapia (*O. niloticus*)/~100 g	*Bacillus* isolate B29 (Related to *Bacillus subtilis*)	pBESOn-CC (P aprE ^a^, AprE SP ^b^)	CC-Chemokine (Nile tilapia)		1 × 10^8^ CFU/kg of feed. Fish were fed *ad libitum* twice daily for 30 days	Increase in immunoglobulin, complement and lysozyme activity. Improved phagocytic activity.	[147]
Zebrafish (*D. rerio*)/50 mg	*L. lactis* ZHY1	pMG36e-usp45-AcmA-AM (P32 ^a^, Usp45 ^b^, *acm*A ^c^)	pili-like protein Amuc_1100		High-fat diet 10^8^ CFU/g. The zebrafish were fed two times a day at 6% of body weight, for 4 weeks	Reduced hepatic steatosis in zebrafish. Downregulated expression of the lipogenesis [peroxisome-proliferator-activated receptors (PPARγ), sterol regulatory element-binding proteins-1c (SREBP-1c), fatty acid synthase (FAS), and acetyl-CoA carboxylase 1 (ACC1)] and lipid transport genes (CD36 and FABP6) in the liver. Reduced serum aspartate aminotransferase (AST) and alanine aminotransferase (ALT) levels. Decrease in expression of tumor necrosis factor (TNF)-α and interleukin (IL)-6 in the liver. Increase in expression of intestinal tight junction (TJ) proteins (TJP1a, claudina, claudin7, claudin7b, claudin11a, claudin12, and claudin15a. Reduced Proteobacteria and Fusobacteria.	[148]
Zebrafish (*D. rerio*)/0.082 ± 0.002 g	*B. subtilis* wt55	pDG364-N-AIO6 (CotC ^a,b,c^)	AiiO-AIO6 (Lactonase)	*A. veronii* Hm091	10^8^ CFU/g feed. Fish were fed at 6% of body weight per day, increased by 1% after a week, for two weeks.	Improved survival rate. Reduced number of invasive *A. veronii* in gut after challenge. Reduced intestinal alkaline phosphatase activity. Reduced expression of nuclear factor kappa-B (NF-κB) and proinflammatory cytokine interleukin-1β (IL-1β). Increase in expression of lysozyme gene.	[149]

^a^ Promoter, ^b^ Signal peptide, ^c^ CWA.

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
