# Peer review of "Importance of Probiotics in Fish Aquaculture: Towards the Identification and Design of Novel Probiotics"

_microorganisms, 2024, doi:10.3390/microorganisms12030626_

Round 1

Reviewer 1 Report

Comments and Suggestions for Authors

This manuscript described the progress of probiotic application in aquaculture, which is beneficial to further research of aqua probiotics. Some minor revision is needed before publication.

1. The part 3 “Immune system in fish aquaculture” looks not closely related to the subject and should be deleted;

2. The style of manuscript should be modified as the requirement of Journal;

3. The weakness of current studies for aqua probiotics should be mentionedï¼›

4. The conclusion should be included.

Comments on the Quality of English Language

Minor editing of English language required.

Author Response

Dear Reviewer and Editor:

Thank you for all your suggestions. All of them have been addressed in this new version of the article contributing to improving its quality. I will go on to explain each modification suggested by yours in the next lines

Reviewer 1:

Q1. The part 3 “Immune system in fish aquaculture” looks not closely related to the subject and should be deleted.

A1: The section “Immune System in fish aquaculture” was removed

Q2. The style of manuscript should be modified as the requirement of Journal;

A2: With the modification suggested by the reviewers we expect to improve the style of the article according to the Journal. For example, we introduce a Conclusion section (line , we remove the immune chapter to be more close to the Microbiological focus of the Journal

Q3. The weakness of current studies for aqua probiotics should be mentioned

A3: Several weaknesses are pointed out in the review, for example:

  1. There is a lack of comprehensive studies in aquaculture concerning the effects of microbial/probiotic products, such as neurotransmitters (lines 311-315).
  2. The number of studies related to recombinant probiotics in fish species of aquaculture interest is limited (lines 378 to 381
  3. There is a shortage of diverse vectors for producing recombinant probiotics without introducing antibiotic resistance genes or plasmids (lines 494 to 496, and lines 558 to 561).
  4. The number of microbial hosts capable of producing recombinant probiotics that can effectively survive the conditions of the fish gastrointestinal tract is limited (lines 501 to 510).
  5. Most studies have focused solely on pathogens that pose significant challenges to aquaculture production. Consequently, few or no alternative probiotics are available for treating several diseases currently affecting diverse fish species produced in aquaculture (lines 539 to 544).
  6. The design or identification of novel probiotics is hindered by the limited or, in some cases, nonexistent knowledge of the mechanisms of pathogenesis and immune responses of each pathogen and aquaculture species, respectively (line 550 to 557).
  7. There are few available thermostable probiotics or technologies capable of overcoming the high temperatures applied during the process of incorporating probiotics into fish food (lines 528 to 533, and 575 to 579).

Q4. The conclusion should be included.

A4: The section was included (lines 582 to 612)

“5. Conclusions

Probiotics in fish aquaculture are a promising alternative to reduce the negative impact of pathogen outbreaks, reducing the economic losses produced by mortality of specimens, and the use of antibiotics applied to control the bacterial pathogen. Such factors will help make fish aquaculture a more environmentally friendly industry. Currently, most of the probiotics tested in fish aquaculture have been previously studied or applied in humans or mammals, making the development of new probiotics specialized for use in fish necessary. To achieve this goal, a better understanding of the mechanisms of interaction between fish intestinal microbiota and the host is necessary, characterizing the microbial metabolites involved that help to reduce the impact of the outbreaks either by immunostimulant or antagonisms with the pathogens. Whole metagenomics studies could assist this characterization, allowing the identification of microorganisms without genes encoding for virulence factors, able to produce these microbial metabolites or those that encode for genes responsible for the synthesis of structural molecules with immunostimulant properties such as a-Gal. Recombinant probiotics are other alternatives that allow the engineering of probiotics with specific immunostimulant, immunization, or metabolic properties, by the expression of genes that encode for these functions. These recombinant probiotics must be engineered using food-grade plasmids or ideally by the insertion of these genes in the chromosome of the bacterial probiotics without the presence of genes encoding for antibiotic resistance, using modern technologies of genetic engineering such as CRISPR-CAS. However, to aid the proper design of recombinant pro-biotics, a better comprehension of the immune response of each species of fish produced in aquaculture against each pathogen (bacterial, fungal, or viral) is necessary to identify specific targets in the immune response of hosts to be stimulated or repressed. On the other hand, an improved understanding of the pathogenesis mechanism will allow the identification of targets in the pathogens to be selected as antigens to be over-expressed in the probiotics. These research lines must be accompanied by the improvement of the technology to include these probiotics in the fish feed for a successful application in the Fish aquaculture industry.”

Q5: Comments on the Quality of English Language “Minor editing of English language required.”

A5: The Grammar, spelling, and scientific style were checked by a scientist whose native language is English

Reviewer 2 Report

Comments and Suggestions for Authors

1. The focus of this review is on the introduction of probiotics, so the content of chapter 3 is very abrupt and strange in the manuscript, which seems to have little significance. Similarly, so is the title. It is suggested to focus on probiotics.

2. L84-86: Generally speaking, Aeromonas and Vibrio are pathogenic bacteria in aquaculture, so it seems inappropriate to list them as probiotics here.

3.L100: In 2018, this share was 52 percent. Can I update the data to the latest year?

4. L268: "Immune system in fish aquaculture". The title seems inappropriate.

5. The Latin scientific names in the text should be abbreviated to generic names when they appear for the second time.

6. As we all know, the role of probiotics is not only its own, but also whether the probiotic function of its metabolites is outstanding. It is suggested to increase the introduction of probiotics metabolites and their application in aquaculture.

Comments on the Quality of English Language

Minor editing of English language required.

Author Response

Dear Reviewers and Editor:

Thank you for all your suggestions. All of them have been addressed in this new version of the article contributing to improving its quality. I will go on to explain each modification suggested by yours in the next lines

Reviewer 2:

Q1. The focus of this review is on the introduction of probiotics, so the content of chapter 3 is very abrupt and strange in the manuscript, which seems to have little significance. Similarly, so is the title. It is suggested to focus on probiotics.

A1: The chapter 3 was removed, to improve the focus on probiotics. The title was modified from:

“Importance of Probiotics and Fish Immunology in Aquaculture: Towards the Identification and Design of Novel Probiotics”

To

“Importance of Probiotics in Fish Aquaculture: Towards the Identification and Design of Novel Probiotics” (lines 2 and 3)

Q2. L84-86: Generally speaking, Aeromonas and Vibrio are pathogenic bacteria in aquaculture, so it seems inappropriate to list them as probiotics here.

A2: The paragraph was modified from

“Among the bacterial genera that have been used most probiotic microorganisms in aquaculture practices we find the group of lactic acid bacteria (LAB), Bacillus, Aeromonas, Alteromonas, Arthrobacter, Bifidobacterium, Clostridium, Paenibacillus, Phaeobacter, Pseudoalteromonas, Pseudomonas, Rhodosporidium, Roseobacter, Streptomyces and Vibrio, and microalgae (Tetraselmis) and yeasts, Debaryomyces, Phaffia and Saccharomyces [13].”

To

“Among the bacterial genera commonly utilized as probiotic microorganisms in aquaculture practices, prominent groups include lactic acid bacteria (LAB), Bacillus, Alteromonas, Arthrobacter, Bifidobacterium, Clostridium, Paenibacillus, Phaeobacter, Pseudoalteromonas, Pseudomonas, Rhodosporidium, Roseobacter, and Streptomyces [12]. Additionally, eukaryotic microorganisms such as microalgae (Tetraselmis) and yeasts from genera Debaryomyces, Phaffia, and Saccharomyces have demonstrated efficacy in probiotic assessments [12]. Furthermore, certain isolates from the pathogenic genera Aeromonas and Vibrio exhibit probiotic properties [13,14].” (lines 81 to 95)

Q3.L100: “In 2018, this share was 52 percent”. Can I update the data to the latest year?

A3: The information was actualized to the latest report published in 2022, but that contains information of 2020. In 2024 is expected that will be published the annual report containing information of 2022. The paragraph was modified from:

“Global aquaculture production has nearly doubled every ten years, demonstrates the significant and growing role of fisheries and aquaculture in providing food, nutrition and employment. At the global level, since 2016, aquaculture has been the main source of fish available for human consumption. In 2018, this share was 52 percent, a figure that can be expected to continue to increase in the long term [6].”

To

“Global aquaculture production has nearly doubled every ten years, demonstrating the significant and growing role of fisheries and aquaculture in providing food, nutrition, and employment. At the global level, since 2016, aquaculture has been the main source of aquatic animals available for human consumption. In 2020, this share was 56%, a figure that can be expected to continue to increase in the long term[20]. In the same year, Fish (finfish) accounted for 76% of the total aquatic animals produced through aquaculture [20].” (lines 98 to 103)

Q4. L268: "Immune system in fish aquaculture". The title seems inappropriate.

A4: The chapter was removed

Q5. The Latin scientific names in the text should be abbreviated to generic names when they appear for the second time.

A5: We carefully identified the first time that the scientific name appeared and modified the text according to the suggestion established by the reviewer.

Q6. As we all know, the role of probiotics is not only its own, but also whether the probiotic function of its metabolites is outstanding. It is suggested to increase the introduction of probiotics metabolites and their application in aquaculture.

A6: We introduced an entirely new section with this topic. (lines 221 to 310)

“2.2 Microbial metabolites produced by probiotics and intestinal microbiota.

Studies using mammalian models and zebrafish have shown that communication between microorganisms (probiotics or microbiota) and the host involves chemical cross-talk[42]. This communication involves interactions between host receptors/targets on immune cells and metabolites produced by microbial metabolism. This interaction alters the expression of immune genes, modifying the fate of some immune cells or the expression of cytokines [42,43].

Several metabolites produced by microorganisms have the ability to modify host cell metabolism and immune responses [44]. The SCFAs formate, acetate, n-propionate, n-butyrate and n-valerate are molecules produced by the fermentative anaerobic metabolism of bacteria belonging to the gut microbiota (mostly Clostridiales, from phylum Firmicutes). They are among the microbial molecules with the most significant impact on host physiology, reaching distant organs such as the brain due to their hydrophobic nature and low size that enables them to be absorbed by intestinal epithelial cells and to diffuse through the host producing effects in distal organs[45]. Butyrate is the most characterized microbial SCFA. It stimulates the extra-thymus production of Treg, PolyMorfoNuclear lymphocyte (PMN) activity, and the maturation and function of microglia[46,47], reduces the production of pro-inflammatory cytokines INF-γ, IL-1β, and TNF-a in macrophages[48], increases apoptosis and reduces the proliferation of T helper lymphocytes[49]. In dendritic cells, butyrate decreases the exposure of MHC-II, stimulating the production of anti-inflammatory cytokines (IL-22 and IL-10)[50,51]. In general, butyrate (and other SCFAs) produces an anti-inflammatory response; however, its precise effect depends on the SCFA and cell type. The wide spectra of effects related to butyrate can be explained by its capacity to stimulate the mammalian G protein-coupled receptor (GPCR), GPR41, GPR43, and GPR109a, beginning a cascade of phosphorylation mediated by ERK1/2 MAP kinase[52,53]. These receptors are differentially expressed in immune cells. For example, GPR43 is highly expressed in monocytes, macrophages/microglia, and neutrophils[46,48]. Butyrate also inhibits histone deacetylase 3 (HDAC3) involved in chromatin remodeling and produces epigenetic changes [54] that modify cell fate of immune cells. In fish, butyrate has been identified in the gut of carnivorous and herbivorous specimens[55,56] and promotes the expression of heat shock protein HSP70, pro-inflammatory factors (IL-1β and TNF-a), and anti-inflammatory cytokines (TGF-β) in Cyprinus carpio[57], and improves the inflammatory response in juvenile zebra fish[58]. Butyrate has been detected in the intestinal feces of Atlantic salmon at a concentration of around 1 mM[59,60]. Butyrate also has an immunostimulant activity when administered orally to Atlantic salmon, increasing the expression of mRNA encoding for C3 (complement marker) in head-kidney[61] by a mechanism currently unknown. Butyrate also inhibits the antiviral response in SHK-1 cells, inducing the expression of IL-10 and TGFβ in a mechanism independent of the expression of the butyrate receptor [59].

The intestinal microorganisms, and also some probiotics, can metabolize the amino acid tryptophan (Trp) to produce indole-containing metabolites that regulate the immune system activating the aryl hydrocarbon receptor (AHR)[62]. The bacteria responsible for this metabolism belong to the Firmicutes phylum, including members of the Lactobacillus, Clostridium, and Bacillus genera[63]. These metabolites stimulate the production of anti-inflammatory cytokines, promoting host-gut microbiota homeostasis[62]. Microbial indole-3-lactic acid (ILA) promotes the differentiation of CD4+ intraepithelial lymphocytes (IELs) into CD4/CD8 double-positive IELs[64]. Indole-3-acetic acid (IAA) and tryptamine (TRA) reduce the expression of inflammatory mediators such as TNF-a and IL-1β on monocytes /macrophages[65]. Indole-3-aldehyde (I3A) increases the expression of IL-22 in pancreatic innate lymphoid cells and promotes their differentiation toward regulatory macrophages and T-reg lymphocytes[66]. Indole-3-propionic acid (IPA), and indoxyl-3-sulfate (I3S) also regulate T cells and dendritic cells (DC) in the CNS[67]. Kynurenine (Kyn) and its derivates are also immune-active molecules that promote the apoptosis of Th1 cells, increasing the expression of IL-22 and a general anti-inflammatory response[66,68]. Some studies in other fish such as Solea senegalensis show that in general, the administration of Trp improves the immune response by reducing the expression of inflammatory cytokines[69]. Trp can mitigate cannibalism, improve the growth of Asian Sea Bass[70], and counteract the effects of acute stress in Atlantic salmon [71]. In the case of Kyn, it has been described as a pheromone in rainbow trout (Oncorhynchus mykiss); however, there are no reports associated with its function as an immunomodulator[72]. Recent metabolomics studies have identified the presence of ILA, IAA, TRA, Kyn and Trp in the intestinal feces of Atlantic salmon[73].

In addition to Trp metabolites, microorganisms can also produce or activate neurotransmitters such as dopamine, norepinephrine, serotonin, gamma-aminobutyric acids (GABA), acetylcholine, and histamine, which have direct effects on immune cells[74,75]. Receptors for dopamine are found in macrophages, dendritic cells, B lymphocytes, T lymphocytes, microglia, neutrophils, and NK cells. Dopamine has been shown to inhibit Treg cells[76] and/or promote their differentiation to Th2 cells[77]. Norepinephrine is recognized by adrenergic receptors (alpha and beta), which are present in various immune cells. In peripheral tissues, norepinephrine interacts with dendritic cells modifying the production of IL-10, IL-12, and IL-33, which in turn induce changes in naive T lymphocyte differentiation, modifying the balance among the T helper lymphocytes Th1, Th2, and Th17[78]. Serotonin produces several immune effects, depending on its concentration and the type of serotonin receptor expressed on the immune cells[79]. Its production in intestinal enterochromaffin cells is stimulated by microbiota metabolites such as SCFAs[80]. GABA produces different effects in the intestine depending on the cell type and the receptor; while in macrophages it promotes an inflammatory state with an increase in IL-1β, in dendritic cells and T lymphocytes it promotes an anti-inflammatory state with an increase in Treg cells[81]. Acetylcholine shows cell-dependent effects. Specifically, in macrophages it stimulates an inflammatory state through the production of IL-6, TNF-a, IFN-g, and IL-12, while in T lymphocytes it promotes the formation of Treg cells[82]. Histamine shows pleiotropic effects that depend on the receptor stimulated. The histamine interaction with the H2R receptor results in an anti-inflammatory state that increases the production of IL-10 and inhibits the differentiation of T lymphocytes; however, its interaction with other histamine receptors produces an inflammatory stage increasing the production of IL-6, and IFN-g, and promoting the differentiation of T lymphocytes to different T lymphocytes (Th1, Th2, and Th17) depending on the histamine receptor stimulated[83]. These neuro-immunomodulators are produced by several bacteria; for example, some bacterial strains from Lactobacillus or Pseudomonas genera can produce dopamine, norepinephrine, serotonin, and histamine[75].

In aquaculture, few studies have analyzed the relationship between neurotransmitters and immunity. Most of the research has been performed on Rainbow trout and shows that serotonin and dopamine are increased in fish infected with F. psychrophilum[84]. Serotonin also reduces the proliferation of T lymphocytes[85], and acetylcholine reduces the expression of inflammatory cytokines in response to Poly I:C[86].

Q7: Minor editing of English language required.

A7: The Grammar, spelling, and scientific style were checked by a scientist whose native language is English

Reviewer 3 Report

Comments and Suggestions for Authors

The submitted manuscript of Torres-Maravilla et al. is a comprehensive review of the scientific achievements regarding the impact of probiotic use on various fish pathogen infections with particular emphasis on species important for aquaculture.

The authors even emphasize in the title that they focused on fish in aquaculture. For this reason alone, the article constitutes valuable material for review and may indicate directions for future research. Unfortunately, the authors did not avoid quite a number of errors that were more editorial than substantive, although there were some in the content. I suggest to all co-authors, especially those dealing with aquaculture and fish diseases, to give more attention to proofreading.

Below are some comments that, in my opinion, may be helpful in improving the manuscript:

1. In the Title and the rest of the manuscript, apart from the Introduction and Table 1, the authors focus on fish, which are vertebrates. Only in the Introduction and Table 1 did they mention Procamburus clarkii, Macrobrachium rosenbergii and Lithopenaeus vannamei, which are invertebrates. I think that a good solution would be to omit this topic of crustaceans, because in the further content the authors will completely omit this information, which makes mentioning it completely pointless. The lack of discussion of the functioning of, for example, the immune system of crustacean means that the informations from the Introduction and Table 1. have no continuation and are only a copy of information from someone else's articles.

2. Line 56-57: Procamburus clarkii is not a crab, it's a crayfish - red swamp crayfish. It is sometimes called a mudbug but never a mudflat crab. For this reason alone, it is worth getting rid of the crustacean topic from this manuscript.

3. Line 83-81: Please rephrase this sentence, it currently suggests that among the types of bacteria there are also yeasts and microalgae.

4. Table 1 and Table 2. Unify the nomenclature of species, provide proper both common and Latin names. Common names, e.g. common carp, should always be written in the same way, usually with a lowercase letter, but not sometimes with a capital letter and sometimes with a lowercase letter. Only when you start a sentence with a common name or, for example, a line in a table, write Common carp. Remove crustaceans from table 1.

5.  In the text of the manuscript, in many places, correct errors in the names of fish and diseases, only individual examples are given below:
- Latin names always in italics and common names in standard font, e.g. line 211 whiteleg schrimp and line 55 carassius sp., Labeo rohita (line 128)
- standardize the common names of fish in the text of the manuscript, e.g. in the sentence tilapia with a lowercase letter (e. g. line 203)
- when using the abbreviation of a disease or pathogen, e.g. in lines 455-457 IPNV and VHSV, its full name should be written in the content for the first time
- do not write with a capital letter in the sentence, e.g. Spring Viremia in Common Carps. The correct name of the disease is "spring viraemia of carp"

6. In Chapter 5, where the authors wrote about perspectives, they mentioned probiotics Bactocell and the company MoBio.
What is it, why was it mentioned and how does it relate to aquaculture and fish? These are proper names belonging to a specific company, so they should be explained or removed.
Any references to support the suggestions the authors made in this chapter?

Author Response

Dear Reviewer and Editor:

Thank you for all your suggestions. All of them have been addressed in this new version of the article contributing to improving its quality. I will go on to explain each modification suggested by yours in the next lines

Reviewer 3:

Q1. In the Title and the rest of the manuscript, apart from the Introduction and Table 1, the authors focus on fish, which are vertebrates. Only in the Introduction and Table 1 did they mention Procamburus clarkii, Macrobrachium rosenbergii and Lithopenaeus vannamei, which are invertebrates. I think that a good solution would be to omit this topic of crustaceans, because in the further content the authors will completely omit this information, which makes mentioning it completely pointless. The lack of discussion of the functioning of, for example, the immune system of crustacean means that the informations from the Introduction and Table 1. have no continuation and are only a copy of information from someone else's articles.

A1: We followed the suggestion of the reviewer. Consequently, text related to the aquaculture of invertebrates was removed in introduction and in the table 1

Q2. Line 56-57: Procamburus clarkii is not a crab, it's a crayfish - red swamp crayfish. It is sometimes called a mudbug but never a mudflat crab. For this reason alone, it is worth getting rid of the crustacean topic from this manuscript.

A2: We remove this paragraph

Q3. Line 83-81: Please rephrase this sentence, it currently suggests that among the types of bacteria there are also yeasts and microalgae.

A3: The paragraph was modified from:

“Among the bacterial genera that have been used most probiotic microorganisms in aquaculture practices we find the group of lactic acid bacteria (LAB), Bacillus, Aeromonas, Alteromonas, Arthrobacter, Bifidobacterium, Clostridium, Paenibacillus, Phaeobacter, Pseudoalteromonas, Pseudomonas, Rhodosporidium, Roseobacter, Streptomyces and Vibrio, and microalgae (Tetraselmis) and yeasts, Debaryomyces, Phaffia and Saccharomyces [13].”

To

“Among the bacterial genera commonly utilized as probiotic microorganisms in aquaculture practices, prominent groups include lactic acid bacteria (LAB), Bacillus, Alteromonas, Arthrobacter, Bifidobacterium, Clostridium, Paenibacillus, Phaeobacter, Pseudoalteromonas, Pseudomonas, Rhodosporidium, Roseobacter, and Streptomyces [12]. Additionally, eukaryotic microorganisms such as microalgae (Tetraselmis) and yeasts from genera Debaryomyces, Phaffia, and Saccharomyces have demonstrated efficacy in probiotic assessments [12]. Furthermore, certain isolates from the pathogenic genera Aeromonas and Vibrio exhibit probiotic properties [13,14].” (lines 81 to 88)

Q4. Table 1 and Table 2. Unify the nomenclature of species, provide proper both common and Latin names. Common names, e.g. common carp, should always be written in the same way, usually with a lowercase letter, but not sometimes with a capital letter and sometimes with a lowercase letter. Only when you start a sentence with a common name or, for example, a line in a table, write Common carp. Remove crustaceans from table 1.

A4: The nomenclature of species in table 1 and 2 were modified and unified according to the reviewer suggestion.

Q5.  In the text of the manuscript, in many places, correct errors in the names of fish and diseases, only individual examples are given below: - Latin names always in italics and common names in standard font, e.g. line 211 whiteleg schrimp and line 55 carassius sp., Labeo rohita (line 128) - standardize the common names of fish in the text of the manuscript, e.g. in the sentence tilapia with a lowercase letter (e. g. line 203) - when using the abbreviation of a disease or pathogen, e.g. in lines 455-457 IPNV and VHSV, its full name should be written in the content for the first time- do not write with a capital letter in the sentence, e.g. Spring Viremia in Common Carps. The correct name of the disease is "spring viraemia of carp"

A5: All modifications suggested were introduced

Q6. In Chapter 5, where the authors wrote about perspectives, they mentioned probiotics Bactocell and the company MoBio. What is it, why was it mentioned and how does it relate to aquaculture and fish? These are proper names belonging to a specific company, so they should be explained or removed. Any references to support the suggestions the authors made in this chapter?

A6a:  The changes suggested by the reviewers were introduced two paragraphs.  In the first paragraph where Bactocell is indicated, the paragraph was modified

From:

“However, their non-specific effect on pathogens, coupled with the difficulties in isolating probiotic microorganisms from fish microbiota, as well as their incorporation into feed through extrusion processes carried out at high temperatures that reduce the number of viable cells, impose a barrier that has only been successfully overcome in aquaculture by probiotics such as Bactocell.”

To

“However, their non-specific effect on pathogens, coupled with the difficulties in isolating probiotic microorganisms from fish microbiota, as well as their incorporation into feed through extrusion processes carried out at high temperatures that reduce the number of viable cells, impose a barrier that has only been successfully overcome in aquaculture by probiotics such as P. acidilactici CNCM I-4622 - MA 18 / 5M (Bactocellâ, Lallemand)[152–154].” (lines 528 to 533)

A6b. The paragraph containing the MoBio information was modified, to remove the name of the company and include a reference. The paragraph was modified from:

“Although companies like MoBio have developed plasmids with metabolic markers for expression in L. lactis, plasmids have the potential to be transferred among related microorganisms, even if they do not possess selection markers.”

To

“Although food-grade plasmids with metabolic markers for expression in L. lactis have been developed by [155], plasmids have the potential to be transferred among related microorganisms, even if they do not possess selection markers.” (lines 561 to 564)

Round 2

Reviewer 3 Report

Comments and Suggestions for Authors

As I wrote in the first review, please correct the nomenclature. In the corrected version of the manuscript, I noticed again a few errors, i.e. in the naming of taxa in Latin and English. There is still a need to standardize species nomenclature and provide appropriate common and Latin names. Please look at this issue carefully again, especially the spelling in lowercase and uppercase letters and in italics. Due to the fact that this is a manuscript in the form of a review and not a description of your own research, special attention should be paid to the correct spelling and avoid repeating errors.
Among others :
- line 177 should be pike perch, not Pike perch
- line 110 should be Carassius sp. or crucian carp and not carassius sp.
- line 190 and Table 1: standardize the names for Labeo rohita, once it says Ruhu, Rohu, once in italics, once in normal font. If the authors do not know how to write it, it is safer to leave only the Latin species name in italics.
- in Table 2, reference no. 143 should be C. carassius and not C. Carassius

Author Response

Dear Reviewer

Thank you very much for your suggestion. We took your advice and improved the quality of the manuscript introducing the following changes.

Line 57

From

“Carassius sp.”

To

“crucian carp (Carassius sp.)”

Line 68

From

“FAO”

To

“food and agriculture organization of the united nations (FAO)”

Line 126

From

“Pike-perch”

To

“pike-perch”

Line 140

From

“Rohu”

To

“rohu”

Line 227

From

“zebrafish”

To

“zebrafish (Danio rerio)”

Line 280 

From

Solea senegalensis”

To

Senegalese sole (Solea senegalensis)”

Line 283

From

“Asian Sea Bass”

To

“Asian Sea Bass (Lates calcarifer)”

Line 403

From

“Crucian”

To

“crucian”

Line 437

From

“xylose or nisin (Table 2)”

To

“xylose (Pxyl) or nisin (Pnis) (Table 2)”

Table 2

From

“pXyl”

To

“Pxyl”

From

“PNis”

To

“Pnis”

A food table was introduced in table 2

“aPromoter, bSignal peptide, cCWA domain”